# Enzymatic Activity as New Moorsh-Forming Process Indicators of Peatlands

**Lech W. Szajdak \*** , **Teresa Meysner** and **Marek Szczepański**

Institute for Agricultural and Forest Environment, Polish Academy of Sciences, Bukowska 19,
60-809 Poznań, Poland; teresa.meysner@isrl.poznan.pl (T.M.); marek.szczepanski@isrl.poznan.pl (M.S.)
\* Correspondence: lech.szajdak@isrl.poznan.pl; Tel.: +48-61-8475-601

**Abstract:** The aim of this study is to comprehensively assess the change in oxidoreductive enzyme activities, due to the potential in catalyzing oxidation and reduction reactions, as the basic processes on undrained and drained peat soils. On undrained peatlands, a significant decrease of enzyme activities was observed such as xanthine oxidase, urate oxidase, phenol oxidase, and peroxidase with an increase in depth. It was connected with significantly higher porosity values, hot water extractable organic carbon, and total organic nitrogen contents, ammonium and nitrate ions concentrations, and significantly lower ash and bulk density values in the upper layers. On drained peatlands, a significant increase of enzyme activities in depth was measured. Enzyme activities such as xanthine, urate, phenol oxidase, and peroxidase were documented to be effective as new indicators and tools for changes of the moorsh-forming process in association with the oscillation of the water table caused by the drainage of the peatlands.

**Keywords:** new moorsh-forming process indicators; oxidoreductive enzyme activities; peatlands; physicochemical properties

## 1. Introduction

Peatlands are a type of wetlands that are the most valuable ecosystems on earth showing an important function in many biogeochemical cycles, through filtering and clarifying water as biogeochemical barriers. In addition, peatlands are critical for preserving global biodiversity, minimizing flood risk, and climate change [1].

Most of the peatland areas are still in a natural state, where only 15% of the world's peatlands have been drained and degraded. It should be noted that peat accumulation depends on the balance between production and decay. Worldwide, undrained peatlands of 3 million km$^2$ presently release up to 100 Megaton of carbon/year [2]. The exact accumulation level depends on the peatland type. Both, too low and high water levels are detrimental to peat accumulation [3].

Adverse peatlands use, especially modification for croplands, pastures, or plantations, including biofuel crops and peat mining is a major example of change in land-use for agriculture and forestry [4]. Peatland degradation as a result of drainage caused subsidence peat oxidation, shrinkage, and compaction. Initial subsidence is caused by compaction and can be more than 50 cm per year and connected with the drainage level, the type, and depth of the peat. After a few years, oxidation is the main factor, causing up to 90% of the subsidence [5].

Where natural peatlands have to be converted into production use, a land-use program should be implemented that is connected with wet conditions (a practice referred to as paludiculture). Ecosystem restoration now focuses on damming or infilling ditches to increase biodiversity. The first step is to raise the water table. In successful restoration, the recolonization of mire plant species follows rewetting, and the carbon cycle begins once again [6].

The intensive peatlands management leads to the lowering of a water table, dehydration and the changes in the air-water conditions in peat. All these processes stimulate the chemical, physical, and biochemical breakdown of organic matter. This means that moorsh cannot return to the original conditions of peat. This process is usually called the secondary transformation of peat [7].

The moorsh-forming process reflects the changes taking place in the structure of the organic mass constituting these soils leading to the changes in the properties of high molecular weight compounds from hydrophilic to hydrophobic, conversion of organic functional groups and disturbances to the state of balance in the colloidal systems of the organic matter [8].

The mechanism of these processes is very complex. All these negative changes lead to the irreversible shrinking of organic matter since secondary bindings are formed among their molecules [9].

The physicochemical property of the peat is one factor and tool that can be assumed to affect the rate of peat oxidation. The main changes in the organic matter are the following:

- Ash content, specific and bulk density increases;
- Porosity, moisture, shrinkage, compressibility, the content of bitumen, cellulose, and lignin is reduced;
- The hydrophilic abilities of moorsh decreases, hydrophobic increases;
- The ratio of fulvic to humic acids, the content of mineral substances and greenhouse gas emissions, in particular $CH_4$, $CO_2$, and $N_2O$ increases.

The first description of the moorsh-forming process was introduced by Okruszko [10,11], who described the morphology of the peat-moorsh soil profile. He characterized surface moorsh horizon-M and the peat horizon-T, dividing moorsh horizon into three layers: M1, M2, and M3.

The evaluation of the degree of the moorsh-forming process was described as follows:

Hooghoudt et al. [9] as the first, attempted to describe drained organic soils quantitatively in terms of their ability to absorb water. Their method compares the water retention capacity of a soil sample taken from the surface layer after drying at 105 °C with the water retention capacity of the sample not dried and the ability to retain water by peat soil from the same profile. In addition, evaluation of the degree of transformation on a 10-point scale was described.

Schmidt [12] developed a determination method of the transformation organic matter state, exposed to drainage, based on the amount of water held at the 100 kPa suction power to the mass of dried soil. The changes of oxic conditions in peatlands are related to some physical, chemical, and biochemical parameters such as the humification index, specific surface area, or enzyme activity.

Gawlik [13] developed a modified version of the volume-weight method. Differentiation between the water holding capacity of peat-moorsh materials was expressed with the help of the $W_1$ index. The results were compared to the secondary transformation classes proposed by the author.

Sokołowska et al. [8] showed a linear relationship between the state of the secondary transformation of peat $W_1$ and the content of hydrophobic amino acids in organic soil colloids. An increase in the content of hydrophobic amino acids was documented during the moorsh-forming process in highly transformed soils.

The literature does not report any comprehensive investigation into the changes of oxidoreductive enzyme activities in relation to physicochemical properties caused by moorsh-forming processes. Commonly used physical methods such as those by Okruszko, Hooghoudt et al., Schmidt, Gawlik, and biochemical ones by Sokołowska et al. do not concern the cause of this process, but describe its effect thus representing a low effectiveness in the evaluation of moorsh-forming processes. Estimating only one peat property, which is highly heterogeneous, makes this method very limited because of the fact that it does not present clear and precise information on the assessment of the moorsh-forming process.

The "anti-effect" of drained peatlands needs complex combined chemical, biochemical, and physical studies to evaluate the mechanism and moorsh-forming processes.

The moorsh-forming process takes place in aerobic conditions and oxidoreductive properties of peat. Therefore oxidoreductive enzymes are one of the key factors in organic matter transformation, decomposition and mineralization, being the main indicator of peat quality due to their quick response and sensitivity to external environmental conditions.

The activity of oxidoreductive enzymes: xanthine oxidase, urate oxidase, phenol oxidase, and peroxidase, directly affect the transformation rates of peat biopolymers into substrates, which are easily available to microorganisms and cultivated plants [14,15].

In connection with the above, the aim of this study is to comprehensively assess the change in the enzyme activities of the oxidoreductases class: xanthine oxidase, phenol oxidase, urate oxidase, and peroxidase, due to the potential in catalyzing oxidation and reduction reactions as basic processes on undrained and drained peatlands, in different climate and soil conditions. We propose to use the activity of these enzymes as a new indicator and tool of the moorsh-forming process of peat. The results may help to form a strategy for the monitoring of the peatlands because their adverse management leads to an increase in the world of the emission of gaseous substances such as $CO_2$, $CH_4$, $N_2O$, and $N_2$ into the atmosphere.

## 2. Materials and Methods

### 2.1. Study Area

For the study, 13 sampling points located on nine peatlands have been chosen (Tables 1 and 2, Figure 1), varying in accordance to their differing types of peat, botanical composition, degree of decomposition, and GPS localizations (determined by TRIMBLE GeoExplorer 3 with accuracy 1–3 m, Sunnyvale, CA, USA).

**Table 1.** Botanical composition of vegetation cover, types of peat, degree of decomposition of undrained peatlands: Mukhrino; Tagan Mire 1; Stążka Mire in 0–50 cm and 50–100 cm layers.

| Place of Sampling | | Botanical Composition of Vegetation Cover of Investigated Places | Layers (cm) | Types of Peat Based on Macrofossil Analysis | Degree of Decomp. (von Post) |
|---|---|---|---|---|---|
| | | Mukhrino | | | |
| Bog peat | KM2 | *Betula nana* L., *Chamaedaphne calyculata* (L.) Moench, *Ledum palustre* L., *Pinus sibirica* Du Tour, *P. sylvestris* L., *Rubus chamaemorus* L., *Sphagnum angustifolium* (Warnst.) C.E.O. Jensen, *S. fuscum* (Schimp.) Klinggr., *S. magellanicum* Brid. | 0–50 | *Sphagnum* | H1 |
| | | | 50–100 | *Sphagnum* | H2 |
| | KM3 | *Betula nana*, *Chamaedaphne calyculata*, *Cladonia sp.* P. Browne, *Ledum palustre*, *Pinus sylvestris*, *Rubus chamaemorus*, *Sphagnum fuscum* | 0–50 | *Sphagnum* | H1 |
| | | | 50–100 | *Sphagnum* | H2 |
| | KM4 | *Chamaedaphne calyculata*, *Ledum palustre*, *Oxycoccus palustris* L., *Pinus sylvestris*, *Rubus chamaemorus*, *Sphagnum fuscum* | 0–50 | *Sphagnum* | H1 |
| | | | 50–100 | *Sphagnum* | H1 |
| | KM10 | *Betula nana*, *Chamaedaphne calyculata*, *Ledum palustre*, *Oxycoccus microcarpus* (Turcz. ex Rupr.) Schmalh., *Pinus sibirica*, *P. sylvestris*, *Sphagnum capillifolium* (Ehrh.) Hedw., *S. angustifolium*, *S. fuscum*, *Vaccinium uliginosum* L., *Vaccinium vitis-idaea* L., *Rubus chamaemorus* | 0–50 | *Sphagnum* | H1 |
| | | | 50–100 | *Sphagnum* | H2 |
| | KM18 | *Andromeda polifolia* L., *Carex limosa* L., *Drosera anglica* Huds., *Eriophorum russeolum* L., *Menyanthes trifoliata* L., *Oxycoccus microcarpus* (Turcz. ex Rupr.) Schmalh., *Rhynchospora alba* (L.) Vahl., *Scheuchzeria palustris* L., *Sphagnum papillosum* Lindb. | 0–50 | *Sphagnum* | H1 |
| | | | 50–100 | *Sphagnum* | H1 |
| | | Mukhrino | | | |
| Fen peat | KM1 | *Aulacomnium palustre* (Hedw.) Schwägr., *Carex globularis* L., *Chamaedaphne calyculata*, *Dicranum polysetum* Sw., *Ledum palustre*, *Oxycoccus microcarpus*, *Pinus sylvestris*, *P. sibirica*, *Pleurozium schreberi* (Willd. ex Brid.) Mitt., *Polytrichum strictum* Brid., *Rubus chamaemorus* L., *Sphagnum fuscum*, *S. capillifolium*, *S. magellanicum*, *S. angustifolium*, *Vaccinium myrtillus*, *Vaccinium vitis-idaea* | 0–50 | sedge woody | H2 |
| | | | 50–100 | woody-cotton grass | H3/H4 |
| | KM15 | *Carex juncea*, *Comarum palustre* L., *Phalaris arundinacea* L., *Lactuca sibirica* (L.) Benth. ex Maxim., *Calamagrostis stricta* (Timm) Koeler *(C. neglecta)*, *C. phragmitoides* Hartm., *Lythrum salicaria* L., *Lysimachia thyrsiflora* L., *L. vulgaris* L., *Rumex aquatilis*, *Galium ruprechtii* Pobed., *Lathyrus palustris* L., *Anemone dichotoma* L., *Betula pubescens* Ehrh., *Salix pentandra* L., *Salix cinerea* L. | 0–50 | sedge woody | H5 |
| | | | 50–100 | sedge woody | H6 |
| | KM16 | *Betula pendula*, *B. pubescens*, *Carex rostrata* Stokes, *C. lasiocarpa* Ehrh., *C. limosa* L., *Eriophorum vaginatum*, *Lysimachia thyrsiflora*, *Menyanthes trifoliata*, *S. riparium* L., | 0–50 | sedge-*Sphagnum* | H2 |
| | | | 50–100 | herbaceous-Equisetum | H2 |

**Table 1.** *Cont.*

| Place of Sampling | Botanical Composition of Vegetation Cover of Investigated Places | Layers (cm) | Types of Peat Based on Macrofossil Analysis | Degree of Decomp. (von Post) |
|---|---|---|---|---|
| | Tagan Mire 1 | | | |
| Fen peat | *Betula pendula, Calamagrostis stricte* (Timm.) Koeler, *Carex lasiocarpa* Ehrh., *Carex sp., Hieracium sp.* L., *Larix sibirica* Ledeb., *Picea obovata* Ledeb., *Pinus cembra var. sibirica* Du Tour, *Pinus sylvestris, Prunus avium* L., *Sorbus aucuparia* L., *Viburnum opulus* L. | 0–50 | grasses | H4 |
| | | 50–100 | grasses | H4 |
| | Stążka Mire | | | |
| Fen peat | *Alnus glutinosa* Gaertn., *Andromeda polifolia, Calluna vulgaris* (L.) Hull, *Carex limosa, Drosera rotundifolia* L., *Empetrum nigrum* L., *Eriophorum vaginatum, Juncus effusus* L., *Ledum palustre, Pinus sylvestris, Pleurozium schreberi* (Willd. ex Brid.) Mitt., *Rhynchospora alba* (L.) Vahl., *Scheuchzeria palustris* (L.) Dulac., *Sphagnum cuspidatum* Ehrh. ex Hoffm., *S. fallax* (Klinggr.) Klinggr., *S. fuscum, S. magellanicum, Vaccinium oxycoccos* L. | 0–50 | sedge-*Hypnum* | H3 |
| | | 50–100 | sedge, fragments of wood | H4/H5 |

**Table 2.** Botanical composition of vegetation cover, species of peat, degree of decomposition of drained peatlands: Wrześnica River valley; General Dezydery Chłapowski Landscape Park 1, 2, 3, 4; Tagan Mire 2; Great Batorowskie; Zieleniec Mire in 0–50 cm and 50–100 cm layers.

| Place of Sampling | Botanical Composition of Vegetation Cover of Investigated Places | Layers (cm) | Types of Peat Based on Macrofossil Analysis | Degree of Decomp. (von Post) |
|---|---|---|---|---|
| | Wrześnica River valley | | | |
| Fen peat | 1   *Achillea millefolium* L., *Alopecurus geniculatus* L., *Caltha palustris* L., *Carex gracilis* L., *C. hirta* L., *Cirsium arvense* (L.) Scop., *C. oleraceum* (L.) Scop., *Crepis paludosa* L., *Daucus carota* L., *Eleocharis palustris* L., *Equisetum fluviatile* L., *Erodium cicutarium* (L.) L'Her., *Filipendula ulmaria* L.) Maxim., *Galium palustre* L., *Heracleum sphondylium* L., *Iris pseudacorus* L., *Juncus articulatus* L., *J. conglomeratus* L., *J. effusus* L., *Lychnis flos-cuculi* (L.) Greuter & Burdet, *Myosotis palustris* L., *Phragmites australis* (Cav.)Trin. ex Steud, *Plantago lanceolata* L., *P. major* L., *Polygonum hydropiper* L., *Potentilla anserina* L., *Prunella vulgaris* L., *Ranunculus acris* L., *R. repens* L., *Rorippa palustris* L., *Rumex acetosa* L., *R. obtusifolius* L., *Scirpus sylvaticus* L., *Stellaria media* (L.) Vill., *Taraxacum officinale* F.H. Wigg., *Trifolium pratense* L., *T. repens, Vicia cracca* L., *Viola palustris* L. | 0–50 | *Alneti* | H5 |
| | | 50–100 | *Alneti* | H2 |
| | 2   *Acorus calamus* L., *Carex acutiformis* L., *C. pseudocyperus* L., *C. rostrata, Cirsium palustre* (L.) Scop., *Epilobium hirsutum* L., *Equisetum palustre* L., *Fallopia convolvulus* (L.) Á. Löve, *Filipendula ulmaria, Galium palustre, Iris pseudacorus* L., *Lycopus europaeus* L., *Lysimachia vulgaris* L., *Lythrum salicaria, Myosoton aquaticum* (L.) Moench, *Ranunculus sceleratus* L., *Rorippa palustris, Rumex hydrolapathum* Huds., *Salix cinerea, S. fragilis* L., *Scirpus sylvaticus* L., *Solanum dulcamara* L., *Symphytum officinale* L., *Urtica dioica* L. | 0–50 | *Alneti* | H5 |
| | | 50–100 | *Alneti* | H2 |
| | General Dezydery Chłapowski Landscape Park | | | |
| Fen peat | 1   *Achillea millefolium* L., *Acorus calamus* L., *Alnus glutinosa* (L.) Gaertn., *Bidens frondosa* L., *Carex acutiformis, Cerastium holosteoides* L., *Cirsium arvense, Conyza canadensis* (L.) Cronquist, *Epilobium hirsutum* L., *Galium mollugo* L., *G. palustre, G. uliginosum* L., *Glechoma hederacea* L., *Holcus lanatus* L., *Iris pseudacorus* L., *Lathyrus palustris* L., *Lemna minor* L., *Lycopus europaeus, Lythrum salicaria, Matricaria maritima* (L.) W. D. J. Koch, *Mentha aquatica* L., *Phalaris arundinacea* L., *Phleum pratense* L., *Phragmites australis, Plantago lanceolata, P. major, Polygonum amphibium* (L.) Delarbre, *Potentilla reptans* L., *Ranunculus repens, Rumex crispus* L., *Sonchus asper* (L.) Hill., *Stachys palustris* L., *Taraxacum officinale, Trifolium hybridum* L., *T. repens, Urtica dioica* | 0–50 | moorsh soil, alder swamp | H8 |
| | | 50–100 | wooden sedge, sedge-reed | H8 |
| | 2   *Achillea millefolia* L., *Agrostis canina* L., *Arrhenatherum elatior* (L.) P. Beauv. ex J. & C. Presl, *Carex acutiformis, C. gracilis, Ceratophyllum demersum* L., *Cirsium arvense, C. oleraceum, Deschampsia caespitosa* (L.) P.B., *Epilobium hirsutum* L., *Galium mollugo, Glechoma hederacea* L., *Heracleum sphondylum* L., *Holcus lanatus, Hydrocharis morsus–ranae* L., *Leucanthemum vulgare* Lam., *Lolium multiflorum* Lam., *Lysimachia vulgaris, Lythrum salicaria, Phragmites australis, Plantago lanceolata, P. major, Ranunculus repens, Rumex acetosa, R. crispus, R. hydrolapathum, Salix alba* L., *S. cinerea* L., *Serratula tinctoria* L., *Solanum dulcamara, Taraxacum officinale, Trifolium pratense* L., *T. repens, Typha angustifolia* L., *Urtica dioica* | 0–50 | moorsh soil, sedge | H7 |
| | | 50–100 | sedge-reed | H8 |
| | 3   *Achillea millefolium* L., *Agrostis canina, Betula pendula, Calystegia sepium* (L.) R.Br, *Cardaminopsis arenosa* (L.) Hayek, *Carex gracilis, C. hirta, Centaurea jacea* L., *Cerastium holosteoides* Fr. em. Hyl., *Cirsium arvense, C. oleraceum, Dactylis glomerata* L., *Daucus carota, Deschampsia caespitosa, Eupatorium cannabinum* L., *Festuca arundinacea* Schreb., *Frangula alnus* Mill., *Galium album* Mill., *G. uliginosum* L., *Holcus lanatus, Hypericum tetrapterum* Fr., *Lycopus europaeus, Lysimachia vulgaris, Mentha aquatica, Molinia caerulea* (L.) Moench, *Nymphaea alba* L., *Phleum pratense* L., *Plantago lanceolata, P. major, Poa pratensis* L., *P. trivialis* L., *Potentilla anserine* L., *Ranunculus repens, Rhamnus catharticus* L., *Rubus plicatus* W. et N., *Salix cinerea, Solanum dulcamara, Sonchus arvensis* L., *Sparganium ramosum* L., *Taraxacum officinale, Typha latifolia, Viburnum opulus* | 0–50 | moorsh soil, sedge with wooden | H8 |
| | | 50–100 | sedge | H7 |

**Table 2.** *Cont.*

| Place of Sampling | Botanical Composition of Vegetation Cover of Investigated Places | Layers (cm) | Types of Peat Based on Macrofossil Analysis | Degree of Decomp. (von Post) |
|---|---|---|---|---|
| 4 | *Achillea millefolium, Alnus glutinosa, Angelica sylvestris* L., *Caltha palustris, Carex acutiformis, C. gracilis, Centaurea jacea, Cerastium holosteoides, Dactylis glomerata, Deschampsia casespitosa, Echinochloa crus–galli* (L.) P.Beauv., *Eupatorium cannabinum, Filipendula ulmaria, Frangula alnus, Galium mollugo, Glechoma hederacea, Heracleum sphondylium, Holcus lanatus, Humulus lupulus* L., *Lolium multiflorum, Lycopus europaeus, Lysimachia vulgaris, Lythrum salicaria, Mentha verticillata* L., *Phleum pratense* L., *Plantago lanceolata, P. major, Polygonum amphibium, P. persicaria* L., *Ranunculus repens, Rorippa palustris, Rumex acetosa, R. obtusifolius, Sonchus arvensis, Stellaria media, Trifolium pratense, T. repens, Vicia cracca* | 0–50 | moorsh soil, alder swamp | H8 |
|  |  | 50–100 | sedge with wooden | H8 |
| Tagan Mire 2 | | | | |
| Peat-moorsh | *Betula pendula, Calamagrostis stricta, Carex lasiocarpa, Carex sp., Hieracium sp., Larix sibirica, Picea obovata, Pinus cembra var. sibirica, P. sylvestris, Prunus avium, Sorbus aucuparia, Thelypteris palustris* Schott, *Viburnum opulus* | 0–50 | wooden | H4 |
|  |  | 50–100 | wooden-grasses | H4 |
| Great Batorowskie | | | | |
| Bog peat | *Betula pubescens, Calamagrostis villosa, Deshampsia flexuosa, Picea abies, Pinus sylvestris, Polytrichum commune* Hedw., *P. formosum, Sphagnum gigensoni, S. magelanicum, S. palustris., Vaccinium myrtylis* | 0–50 | *Eriophoro-Sphagneti* | H6 |
|  |  | 50–100 | *Eriophoro-Sphagneti* | H5 |
| Zieleniec Mire | | | | |
| Fen peat | *Caluna sp., Betula pubescens, Pinus sylvestris, Sphagnum palustris, Vaccinium myrtylius, V. uliginosum* | 0–50 | *Eriophoro-Sphagneti* | H6 |
|  |  | 50–100 | *Eriophoro-Sphagneti* | H4 |

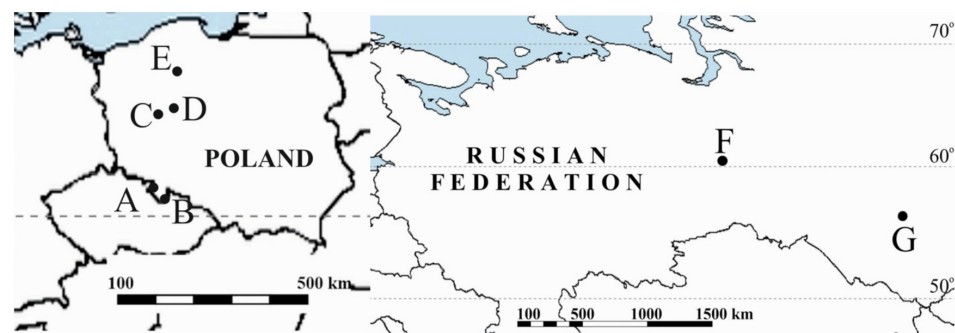

**Figure 1.** Location map of study peatlands. A: Great Batorowskie Peat Bog; B: Zieleniec Mire; C: General Dezydery Chłapowski Landscape Park; D: Wrześnica River valley; E: Stążka Mire; F: Mukhrino; G: Tagan Mire.

### 2.1.1. Undrained Peatlands
#### Mukhrino

The Mukhrino peatland is located in the central Taiga of the West Siberia region (Russia), 26 km west of the town of Khanty-Mansiysk (GPS location: KM1-60°53′41.6″ N, 68°41′51.9″ E; KM2-60°89′50.5″ N, 68°69′20.9″ E; KM3 60°53′43.88″ N, 68°40′46.84″ E; KM4 60°53′44.22″ N, 68°40′12.47″ E; KM10 60°53′29.12″ N, 68°41′33.50″ E; KM15 60°53′55.2″ N, 68°44′59.9″ E; KM16 60°52′35.9″ N, 68°36′46.7″ E; KM18 60°53′43.5″ N, 68°38′20.4″ E) (Table 1, Figure 1).

The water table during summer can reach depths between 5 and 20 cm below the moss surface in poor fens but up to 40–80 cm below the surface in *Sphagnum* hummocks (83–85). The environmental conditions of the Mukhrino peatland are comparable to the subarctic zone of northern Europe.

#### Tagan Mire 1

The Tagan Mire 1 is located 20 km from Tomsk, a city in West Siberia in Russia (GPS location: 56°21′0.0″ N, 84°47′0.0″ E) (Table 1, Figure 1). This peatland is situated on the second flood-plain terrace of the River Tom. The subsoil consists of sand, more seldom

from loamy sand and loam. Eutrophic vegetation such as woody sedge, sedge, sedge-moss, and grass undershrub phytocoenoses covers almost the whole peatland [16].

Stążka Mire

　　　　The Stążka Mire as the Baltic-type fen is located in northern Poland, in the region of the Tuchola Forest on the outwash plain of the Brda River (GPS location: 53°36′17.58″ N, 17°57′20.38″ E (Table 1, Figure 1). This fen is under protection and covers the southern part of the Tuchola Forest National Park and part of the Stążka River Mires Reserve (Table 1, Figure 1). There are spring mires at the channel margins [17].

　　　　The age of undrained peatlands, thickness of the peat deposit, the mean annual air temperature and precipitation are shown in Table 3.

**Table 3.** Age (AD/BC) peatlands, thickness of the peat deposit, mean annual air temperature and precipitation, pH of undrained peatlands: Mukhrino; Tagan Mire 1; Stążka Mire and drained peatlands: Wrześnica River valley; General Dezydery Chłapowski Landscape Park 1, 2, 3, 4; Tagan Mire 2; Great Batorowskie; Zieleniec Mire.

| Sampling Sites | | Age (AD/BC) | Thickness of the Peat Deposit (m) | Mean Annual Air Temperature (°C) | Mean Annual Precipitation (mm) | pH (1M KCl) | |
|---|---|---|---|---|---|---|---|
| | | | | | | Layers (cm) | |
| | | | | | | 0–50 | 50–100 |
| Undrained peatlands | | | | | | | |
| Mukhrino | Bog peat Fen peat | AD 712–780 | 2.0–4.5 | −1.1 | 531 | 2.40–3.28 3.72–4.43 | 2.41–2.77 3.75–4.33 |
| Tagan Mire 1 | | BC 4594–3979 | 9.3 | 0.8 | 532 | 4.83–5.21 | 4.71–5.32 |
| Stążka | | AD 655–779 | 1.4 | 7.2 | 589 | 5.61–5.79 | 7.04–7.26 |
| Drained peatlands | | | | | | | |
| Wrześnica River valley | 1 2 | BC549–398 | 1.4 | 8.9 | 530 | 5.54–6.44 5.52–6.31 | 5.64–6.14 5.49–6.02 |
| General Dezydery Chłapowski Landscape Park | 1 2 3 4 | - | 1.5–2.8 | 7.0 | 650 | 5.13–5.30 5.78–5.91 6.28–6.64 4.63–5.29 | 5.69–5.91 5.81–5.83 5.71–6.30 5.86–6.09 |
| Tagan Mire 2 | | BC 4594–3979 | 9.3 | 0.8 | 532 | 5.14–5.42 | 5.81–5.95 |
| Great Batorowskie | | BC 4225–3961 | 1.2 | 4.8 | 750–920 | 2.50–2.68 | 2.64–2.85 |
| Zieleniec Mire | | BC 166–AD 20 | 6.0 | 6.4 | 665 | 2.75–2.92 | 2.75–2.92 |

2.1.2. Drained Peatlands

Wrześnica River Valley

　　　　Peat-moorsh samples were collected from two sites (at a distance of about 100 m from each other) along the Wrześnica River valley in Goranin of west-central Poland (39 km east of Poznań, Czerniejewo administrative district) (GPS location: 1—52°27′28.37″ N, 17°30′20.96″ E, 2—52°27′28.37″ N, 17°30′20.96″ E) (Table 2, Figure 1).

　　　　The site 1 is an anthropogenic, wet meadow, bogginess, in which the structure of the phytocenosis is formed by 39 species of plants with the domination of the *Scirpetum sylvatici*, *Carex gracilis*, and *Juncus articulatus* communities.

　　　　In the site 2, *Caricetum riparian* community covers the banks of the watercourse as well as the small surface of the river valley where phytocenosis creates 24 species of plants. The dominant species, *Carex rostrata* is one of the peat-forming communities, producing heavy and high-ash sedge peat.

　　　　The soil material represented medium moorsh soil, formed from alder swamp forest peat, strongly decomposed, underlain with lime gyttja.

General Dezydery Chłapowski Landscape Park

The study sites: 1, 2, 3, and 4 are located in the General Dezydery Chłapowski Landscape Park in Turew (40 km south-west of Poznań, West Polish Lowland) on the 4.5-km long transect (GPS location: 1—52°0′57.50″ N, 16°53′49.75″ E; 2—52°1′12.61″ N, 16°53′23.38″ E, 3—52°1′35.45″ N, 16°52′34.80″ E, 4—52°2′21.70″ N, 16°51′9.50″ E) (Table 2, Figure 1). Peat-moorsh soils were classified in line with the Polish hydrogenic soil classification [18], formed from alder swamp forest peat, sedge-reed, sedge, and sedge with wooden peat, underlain with calcareous gyttja.

Tagan Mire 2

Both, Tagan Mire 2, similarly to Tagan Mire 1 are located in West Siberia in Russia (GPS location: 56°21′0.0″ N, 84°48′0.0″ E) (Table 2, Figure 1). The part of the area is drained and under agriculture uses. The subsoil is sandy, and vegetation is represented by woody sedge, sedge, sedge-moss, and grass undershrub phytocoenoses [16].

The Sudetes Peatlands

The Great Batorowskie Bog Peat is located in the Stołowe Mountains National Park (6340 ha) in the Central Sudetes (southwest Poland) (GPS location: 50°27′28.56″ N, 16°23′14.07″ E) (Table 2, Figure 1). The development of organic soils has mainly been influenced by various hydrological conditions [19]. At the turn of the 19th and 20th centuries, peatland ecosystems in the Stołowe Mountains were drained to obtain a new area for afforestation by spruce monoculture. Due to artificial drainage, a rapid drop in the water table was observed [20].

The Zieleniec Mire is located in the Kłodzka Valley, south of Duszniki Zdrój, ranked among the largest raised bog of the Sudety Mountains (GPS location: 50°20′48.64″ N, 16°24′41.92″ E) (Table 2, Figure 1). This part of the Zieleniec Mire was intensively drained at the turn of the 19th and 20th centuries.

The age of drained peatlands, thickness of the peat deposit, the mean annual air temperature and precipitation are shown in Table 3.

### 2.2. Soil Physicochemical and Biochemical Analyses

Peat soil samples were collected from 2014 to 2018 in triplicate from the abovementioned research areas at the depth of 0–50 cm (acrotelm) and 50–100 cm (catotelm), using a 5.0-cm diameter Instorf peat auger in the stratigraphic profile of each peat deposit. The samples were transported to the laboratory at ca. 4 °C and stored at −20 °C. Moreover, a part of soil after removal of roots, stones, etc., was dried at 20 °C and then sieved through a 1-mm sieve. The degree of peat decomposition was estimated by von Post's method [21]. The botanical composition of peat was analyzed microscopically and subsequently classified according to the Polish standards [22] and peat type was described by Lamentowicz et al. [17]. Radiocarbon $^{14}$C dating was performed by accelerator mass spectrometry (1.5 SDH-Pelletron Model "Compact Carbon AMS" (National Electrostatics Corporation, Middleton, USA) and carried out at the Poznan Radiocarbon Laboratory in Poland.

Soil pH values were measured potentiometrically in 1 M KCl (1:2.5 *v/v*) and hygroscopic humidity on moisture analyzer MAX series assayed (Radwag, Radom, Poland). Ash content was determined by placing air-dried peat in a muffle furnace at 550 °C until constant weight. Ash content was expressed as the percentage to the quantity of dry matter. Peat bulk density was estimated from loss-on-ignition values [23] and total porosity calculated from the bulk density ratio of the soil to the density of solids [24].

Hot water extractable organic carbon ($C_{HWE}$) contents were determined using the TOC 5050A analyzer (Shimadzu, Kyoto, Japan). In the investigation of $C_{HWE}$ air-dried samples were heated with deionized water at 100 °C for two hours under a reflux condenser. Extracts were separated by 0.45-µm pore-size filters [25].

The total organic carbon (TOC) contents were measured by inserting about 50 mg of a soil sample in a total organic carbon analyzer (TOC 5050A) with solid sample module (SSM-5000A, Shimadzu, Kyoto, Japan) [26].

The total nitrogen (TN) concentrations were determined by the Kjeldahl method, using the Vapodest 10s analyzer (Gerhardt, Königswinter, Germany). The sample was digested prior to analysis in the presence of $H_2SO_4$, $K_2SO_4$, metallic zinc, and selenium mixture at a final temperature of 350 °C. In the organic matter, some nitrates were present, which were lost during digestion. Free ammonia and organic nitrogen compounds were converted to $(NH_4)_2SO_4$ under these conditions. The ammonia ions concentrations were determined by distillation with 30% NaOH and phenolphthalein in the presence of 4% $H_3BO_3$. The excess of 0.02 N HCl was titrated against standard NaOH using an indicator (bromocresol green and methyl red mixture) that gave a measure of the total nitrogen concentrations of the sample. The endpoint was determined by a change of color from green to red [26].

Ammonium ions ($N-NH_4^+$) concentrations were estimated on an ion chromatograph Waters 1515 (Waters, Milford, MA, USA) appointed with a 1515 Isocratic HPLC pump, conductivity detector Waters 432, a rotary valve $20 \cdot 10^{-6}$ $dm^3$, sample loop, and column PRP-X200 (150 × 4.1 mm I.D.—Internal Diameter) from Hamilton, protected with a guard column (25 × 2.3 mm I.D.) [27].

Nitrate ions ($N-NO_3^-$) concentrations were measured on an ion chromatograph HIC-6A Shimadzu (Shimadzu, Kyoto, Japan) appointed with a LP-6A Isocratic HPLC pump, conductivity detector CDD-6A, a rotary valve with $20 \cdot 10^{-6}$ $dm^3$ sample loop, and column PRP-X100 (150 × 4.1 mm I.D.) from Hamilton, protected with a guard column (25 × 2.3 mm I.D.) [27].

Xanthine oxidase activity (XOA) (EC 1.17.3.2) was determined by the Krawczyński method [28]. Xanthine was used as a substrate for the measurement of xanthine oxidase activity in fresh peat samples. Total of 0.25 g soil was extracted with 50 mL of 0.1 M phosphate buffer (pH = 7.5) and incubated for 4 h at 37 °C. The substrate was filtered and centrifuged at 4000 r.p.m. for 10 min. The mixture consisted of 2.5 mL of the test solution and 2.0 mL of the substrate solution (6.6 mM xanthine). It was incubated for 30 min at 37 °C. The reaction was stopped by the addition of 0.5 mL 0.58 M HCl. A blank was prepared in the same way, but in the place of 2.0 mL 0.1 M phosphate buffer (pH = 7.5), xanthine solution was added. The absorbance was measured at λ = 290 nm.

Urate oxidase activity (UOA) (EC 1.7.3.3.) was determined by the Martin-Smith method [29]. About 5 g peat soil was placed in 250 mL conical flasks, and 100 mL of 0.05 M phosphate buffer (pH = 7.5), 5 drops of toluene was added. The mixture was incubated for 16 h at 24 °C, then filtered and centrifuged at 4000 r.p.m. for 20 min. Next, 4 mL substrate was incubated with 2 mL uric acid for 24 h at 24 °C. A blank was prepared in the same way, but in the place of phosphate buffer, deionized water was added. The absorbance was determined at λ = 293 nm.

Phenol oxidase activity (POA) (EC 1.14.18.1) was determined by the Perucci method [30]. 0.1 M phosphate (pH = 6.5) with 0.2 M of catechol, was oxygenated for 3 min and incubated for 10 min at 30 °C. Then, 3 mL of reagent solution and 2 mL of phosphate buffer to 1.0 g of fresh peat were added. The mixture was incubated for 20 min at 30 °C and the reaction was stopped by cooling in an ice-bath and adding 5 mL of ethanol. Then, this was centrifuged at 4000 r.p.m. for 10 min and filtered. A blank was prepared without soil and catechol. The absorbance of the supernatant fraction was determined at λ = 525 nm.

Peroxidase activity (PA) (EC 1.11.1.7) was determined by the Bartha and Bordeleau method [31]. Total of 100 mL of 0.05 M phosphate buffer (pH = 6) was added to 20 g of fresh peat soil and incubated on a rotary shaker for 1 h at 25 °C. The supernatant was centrifuged at 4000 r.p.m. for 20 min. Then, 3 mL of substrate, 0.5 mL of 0.06% $H_2O_2$ in 0.05 M phosphate buffer (pH = 6), 0.1 mL and 0.5% o-dianisidine in methanol was combined in a 1 cm spectrophotometric cuvette. Dry soil samples and deionized water were used as a blank. The absorbance of the supernatant fraction was measured at λ = 460 nm.

Absorbances for all enzymes were performed colorimetrically on spectrophotometer UV-mini 1240 (Shimadzu, Kyoto, Japan). The activity of enzymes in peat samples was calculated from the early prepared analytical curve according to the Beer–Walter light absorption law by means of the least-squares formulae.

The physicochemical properties and the enzyme activities were measured during the entire vegetation season, and all data have been converted into ash contents and moisture values.

### 2.3. Statistical Analysis

The results were statistically processed using the PQStat 1.8 software (PQStat Software, Poznań, Poland). The correlations between the variables were assessed using Pearson's correlation coefficients at the significance level of $p \leq 0.05$. Data were analyzed using one-way ANOVA for significance level $\alpha = 0.05$.

To determine whether the differences between the two means with the depth (0–50, 50–100 cm) are statistically significant, the p-value was compared to the significance level to assess the null hypothesis. The null hypothesis states that the means are equal ($H_0$: $\mu_1 = \mu_2$). The alternative hypothesis ($H_1$: $\mu_1 \neq \mu_2$) states that the means are not equal. If the *p*-value was less than or equal to the alpha ($p \leq 0.05$), then we rejected the null hypothesis and we said the result was statistically significant. If the *p*-value was greater than alpha ($p > 0.05$), then we failed to reject the null hypothesis, and we said that the result was statistically nonsignificant.

The correlations between the parameters were estimated using Pearson's correlation coefficients at the significance level of $p \leq 0.05$.

## 3. Results

### 3.1. Soil Physicochemical Properties

Peat pH values were found from strongly acidic 2.40 to acidic 5.79 in the upper layer and from strongly acidic 2.41 to neutral 7.26 in the bottom layer on undrained peatlands (Table 3 and Table S1). On drained peatlands, pH values differentiation in the soil profile was from strongly acidic 2.50 to slightly 6.64 of the 0–50 cm layer, and from 2.64 to 6.30 in depth of peat profile (Table S2).

On undrained peatlands, there were no significant differences in the moisture values in depth (Figure 2A, $p > 0.05$). Whereas the study has shown that the soil moisture values of drained peatlands significantly increased with an increase in depth (Figure 2B, $p < 0.05$). The highest increase of the moisture values on drained peatlands was revealed in site 2 of the Wrześnica River valley (*Alneti*) and the lowest in site 4 of the General Dezydery Chłapowski Landscape Park (moorsh soil, alder swamp, sedge with wooden peat) (Table S2).

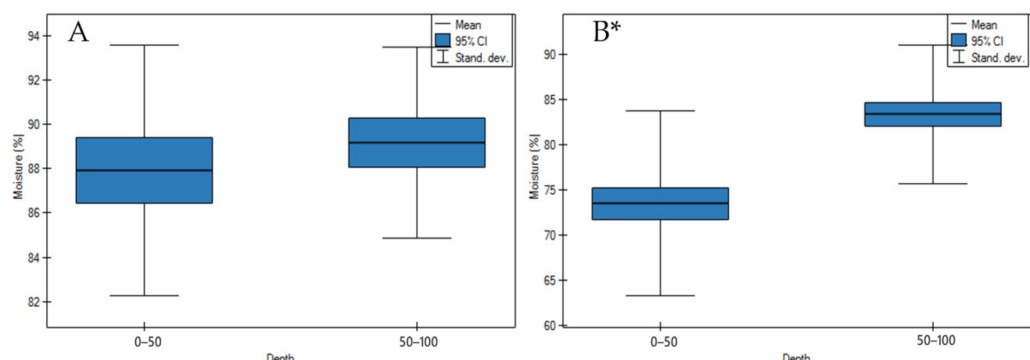

**Figure 2.** Moisture values on: (**A**) undrained, (**B**) drained peatlands in 0–50 cm and 50–100 cm layers. * indicates statistically significant differences ($p < 0.05$) as revealed by ANOVA. (For all cases $n = 60$).

On undrained peatlands, ash contents and bulk density values significantly increased in depth (Figures 3A and 4A, $p < 0.05$). While drainage of the peatlands significantly

impacted on the significant decrease of ash contents and bulk density values in depth (Figures 3B and 4B, $p < 0.05$).

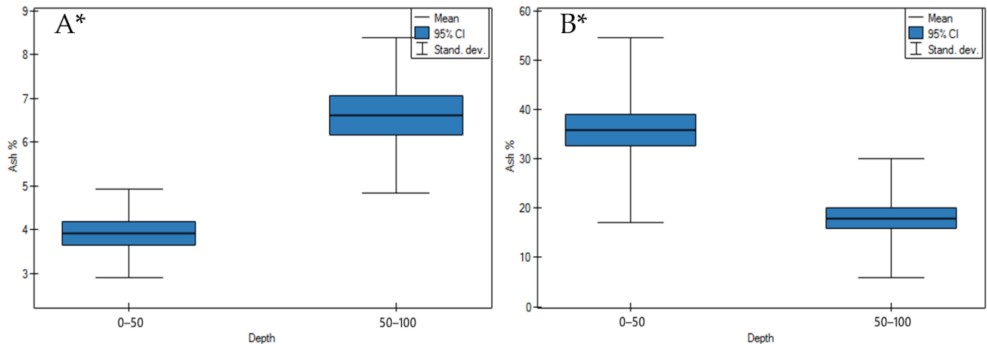

**Figure 3.** Ash values on: (**A**) undrained, (**B**) peatlands in 0–50 cm and 50–100 cm layers. * indicates statistically significant differences ($p < 0.05$) as revealed by ANOVA. (For all cases $n = 60$).

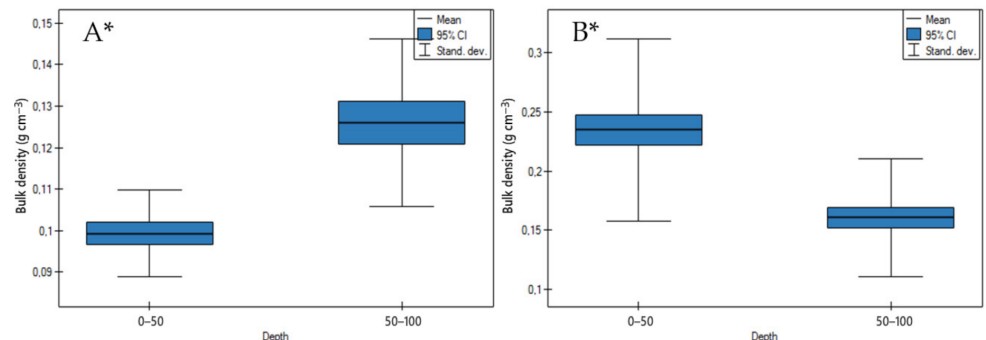

**Figure 4.** Bulk density values on: (**A**) undrained, (**B**) drained peatlands in 0–50 cm and 50–100 cm layers. * indicates statistically significant differences ($p < 0.05$) as revealed by ANOVA. (For all cases $n = 60$).

The porosity values were differentiated and depended on the bulk density values. On undrained peatlands, there were no significant differences in the porosity values in depth (Figure 5A, $p > 0.05$). However, drainage of the peatlands impacted on significant increase in depth of porosity values (Figure 5B, $p < 0.05$).

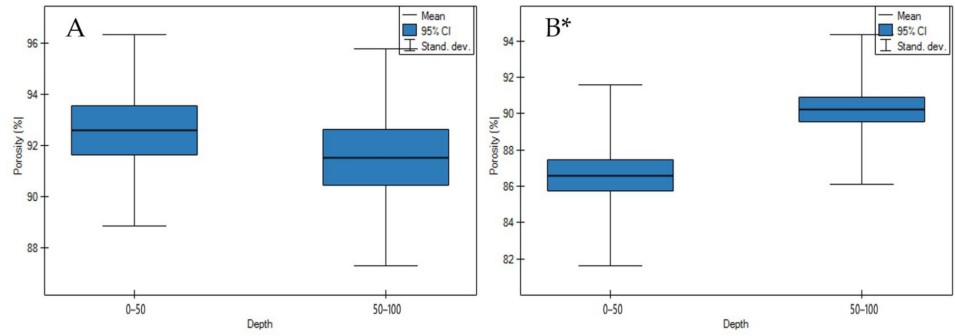

**Figure 5.** Porosity values on: (**A**) undrained, (**B**) drained peatlands in 0–50 cm and 50–100 cm layers. * indicates statistically significant differences ($p < 0.05$) as revealed by ANOVA. (For all cases $n = 60$).

On both undrained and drained peatlands, $C_{HWE}$ contents significantly decreased in depth (Figure 6A,B, $p < 0.05$). On undrained peatlands, the highest decrease of $C_{HWE}$ was demonstrated for the Stążka Mire of Baltic-type (sedge-*Hypnum*, sedge, fragments of wood) and the lowest for the Mukhrino bog peat (*Sphagnum*) (Table S1).

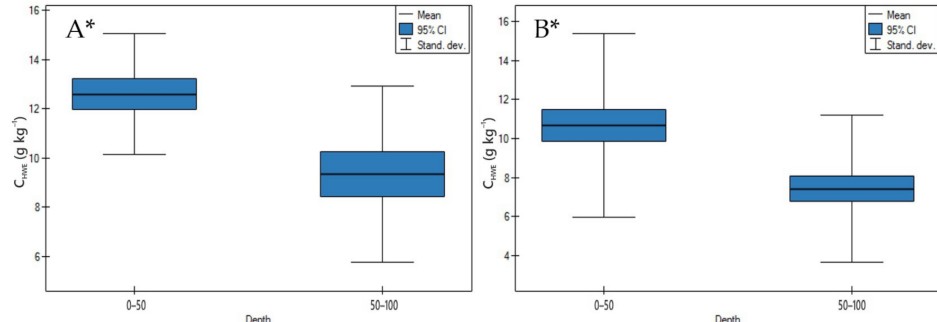

**Figure 6.** $C_{HWE}$ contents on: (**A**) undrained, (**B**) drained peatlands in 0–50 cm and 50–100 cm layers. * indicates statistically significant differences ($p < 0.05$) as revealed by ANOVA. (For all cases $n = 60$).

On drained peatlands, the highest decrease of $C_{HWE}$ contents was measured in site 4 of the General Dezydery Chłapowski Landscape Park (moorsh soil, alder swamp, sedge with wooden peat), and the lowest in site 2 of the Wrześnica River valley (*Alneti*) (Table S1).

On undrained and drained peatlands, TOC contents significantly increased with an increase in depth (Figure 7A,B, $p < 0.05$). On undrained peatlands, the highest increase of TOC contents was measured for the Mukhrino fen peat (sedge woody, woody-cotton grass, sedge-*Sphagnum*, herbaceous-*Equisetum*), and the lowest for the Stążka Mire of Baltic-type (sedge-*Hypnum*, sedge, fragments of wood) (Table S1). On drained peatlands, the highest increase of TOC contents was shown in site 2 of the Wrześnica River valley (*Alneti*), while the lowest for the Great Batorowskie and the Zieleniec Mire (*Eriophoro-Sphagneti*) (Table S2).

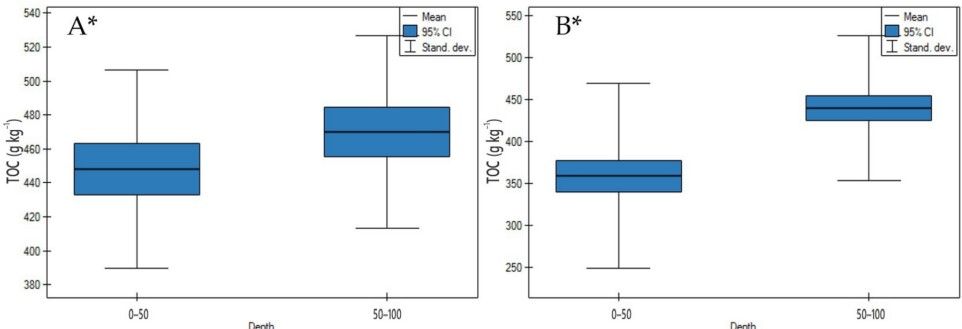

**Figure 7.** TOC contents on: (**A**) undrained, (**B**) drained peatlands in 0–50 cm and 50–100 cm layers. * indicates statistically significant differences ($p < 0.05$) as revealed by ANOVA. (For all cases $n = 60$).

On undrained peatlands, in depth, a significant decrease of TN concentrations was observed (Figure 8A, $p < 0.05$). While on drained peatlands a significant increase of TN concentrations in depth was shown (Figure 8B, $p < 0.05$).

N-NH$_4^+$ concentrations on undrained peatlands significantly decreased with an increase in depth (Figure 9A, $p < 0.05$). The highest decrease in these ion concentrations was observed for the Mukhrino bog peat (*Sphagnum*). In contrast to that on drained peatlands, N-NH$_4^+$ concentrations significantly increased in depth (Figure 9B, $p < 0.05$).

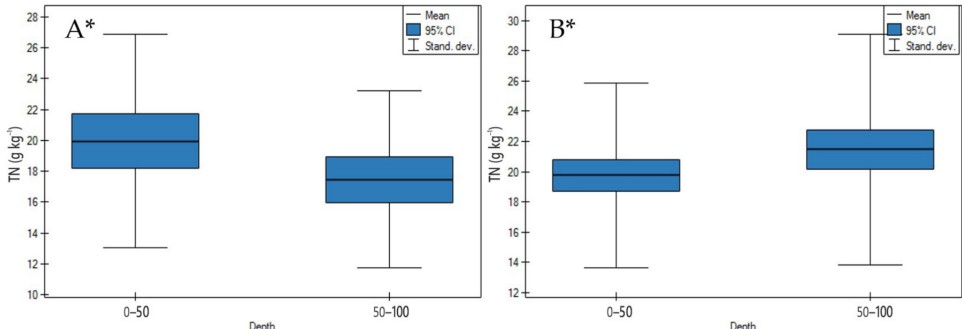

**Figure 8.** TN concentrations on: (**A**) undrained, (**B**) drained peatlands in 0–50 cm and 50–100 cm layers. * indicates statistically significant differences ($p < 0.05$) as revealed by ANOVA. (For all cases $n = 60$).

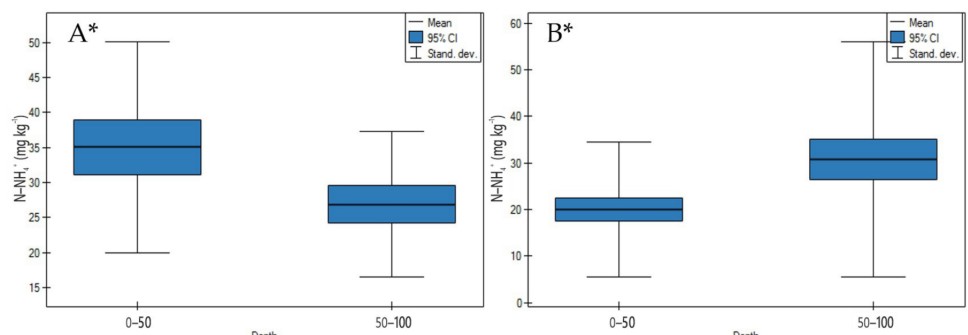

**Figure 9.** N-NH$_4^+$ concentrations on: (**A**) undrained, (**B**) drained peatlands in 0–50 cm and 50–100 cm layers. * indicates statistically significant differences ($p < 0.05$) as revealed by ANOVA. (For all cases $n = 60$).

On undrained peatlands, N-NO$_3^-$ concentrations significantly decreased in depth (Figure 10A, $p < 0.05$). Whereas, on drained peatlands, there was no significant difference between these ions concentration in depth (Figure 10B, $p > 0.05$).

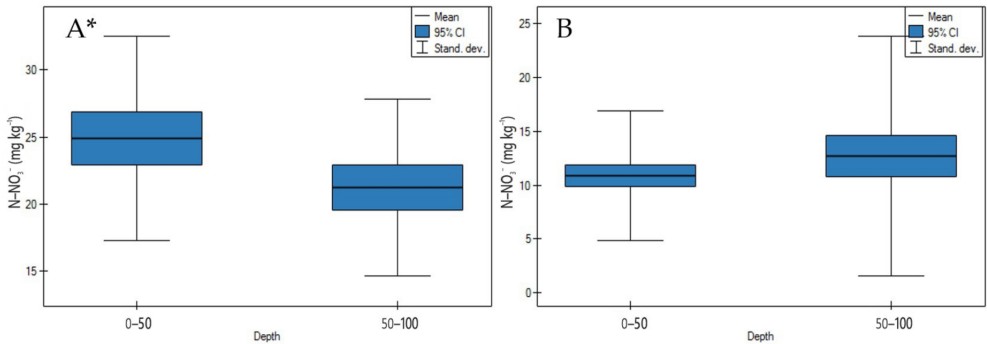

**Figure 10.** N-NO$_3^-$ concentrations on: (**A**) undrained, (**B**) drained peatlands in 0–50 cm and 50–100 cm layers. * indicates statistically significant differences ($p < 0.05$) as revealed by ANOVA. (For all cases $n = 60$).

On both undrained and drained peatlands, the C/N ratio significantly increased in depth (Figure 11A,B, $p < 0.05$). The highest increase of the C/N ratio on undrained peatlands was observed for the Mukhrino fen peat (sedge woody, woody-cotton grass, sedge-*Sphagnum*, herbaceous-*Equisetum*) and the lowest for the Stążka Mire of Baltic-type (sedge-*Hypnum*, sedge, fragments of wood) (Table S1). Moreover, the highest rise of the

C/N ratio in depth on drained peatlands in site 1 of the Września River valley (*Alneti*) and the lowest for the Tagan 2 site (wooden, wooden-grasses) was documented (Table S2).

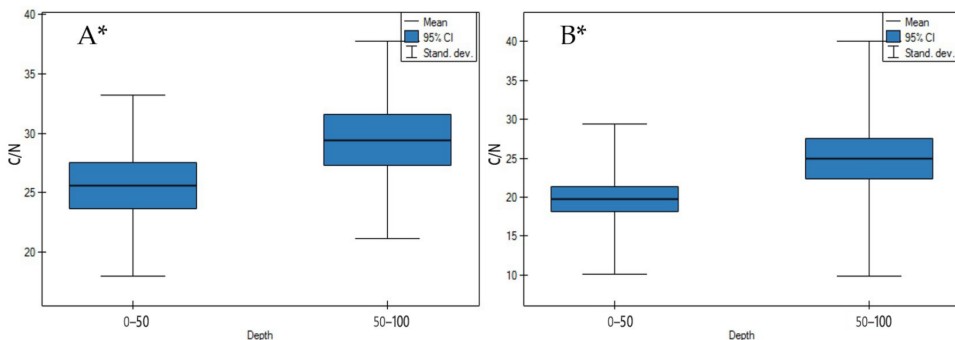

**Figure 11.** C/N ratio on: (**A**) undrained, (**B**) drained peatlands in 0–50 cm and 50–100 cm layers. * indicates statistically significant differences ($p < 0.05$) as revealed by ANOVA. (For all cases $n = 60$).

*3.2. Soil Biochemical Properties*

3.2.1. Xanthine Oxidase Activity

According to Zeng et al. [32], xanthine oxidase is involved in the conversion of hypoxanthine to xanthine and uric acid.

On undrained peatlands with an increase in depth, xanthine oxidase activity significantly decreased (Figure 12A, $p < 0.05$). The highest decrease of enzyme activity was found for the Stążka Mire of Baltic-type (sedge-*Hypnum*, sedge, fragments of wood) and the lowest for the Mukhrino fen peat (sedge woody, woody-cotton grass, sedge-*Sphagnum*, herbaceous-*Equisetum*) (Table S3).

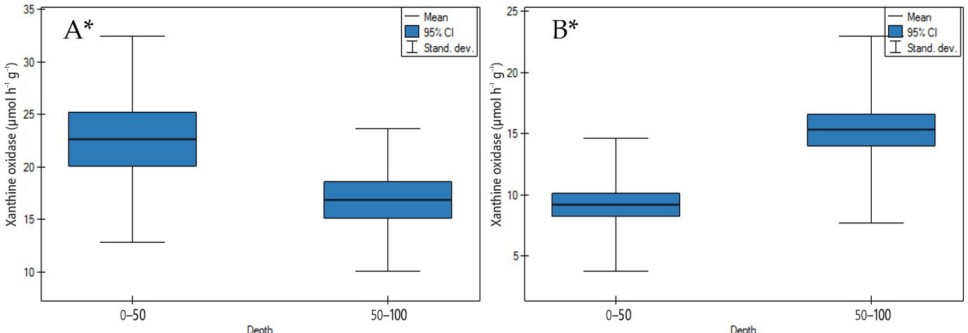

**Figure 12.** Xanthine oxidase activity on: (**A**) undrained, (**B**) drained peatlands in 0–50 cm and 50–100 cm layers. * indicates statistically significant differences ($p < 0.05$) as revealed by ANOVA. (For all cases $n = 60$).

While the drainage of the peatlands led to a significant increase in enzyme activity (Figure 12B, $p < 0.05$). The following points of drained peatlands showed the highest increase of xanthine oxidase activity in site 2 of the General Dezydery Chłapowski Landscape Park (moorsh soil, sedge, sedge-reed), and the lowest in site 2 of the Września River valley (*Alneti*) (Table S4).

3.2.2. Urate Oxidase Activity

The enzyme catalyzes the oxidation of uric acid to allantoin and hydrogen peroxide [33].

On undrained peatlands with an increase in depth, urate oxidase activity significantly decreased for the Stążka Mire of Baltic-type (sedge-*Hypnum*, sedge, fragments of wood) (Figure 13A, $p < 0.05$) (Table S3).

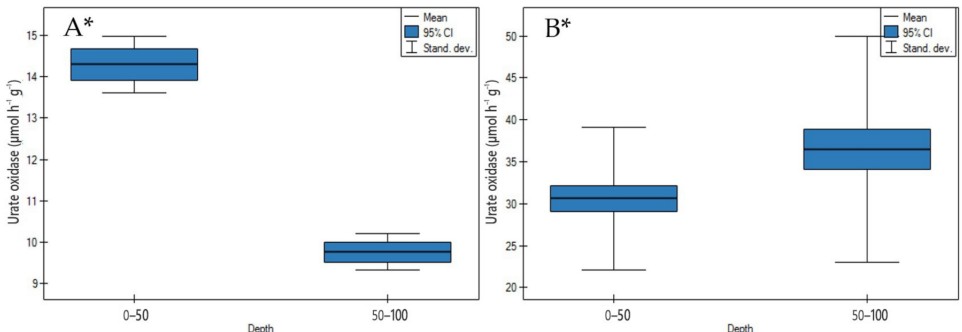

**Figure 13.** Urate oxidase activity on: (**A**) undrained, (**B**) drained peatlands in 0–50 cm and 50–100 cm layers. * indicates statistically significant differences ($p < 0.05$) as revealed by ANOVA. (For all cases $n = 60$).

In contrast to undrained peatlands, the drainage of the peatlands impacted on the significant increase of enzyme activity in depth (Figure 13B, $p < 0.05$). The highest rise of urate oxidase activity was found in site 4 of the General Dezydery Chłapowski Landscape Park (moorsh soil, alder swamp, sedge with wooden), and the lowest in site 2 of the Września River valley (*Alneti*) (Table S4).

### 3.2.3. Phenol Oxidase Activity

The enzyme catalyzes the oxidation of polyphenols to quinones [34] and indicates the central role of phenol oxidase in the $C_{HWE}$ mobilization [35].

On undrained peatlands, phenol oxidase activity significantly decreased with an increase in depth (Figure 14A, $p < 0.05$).

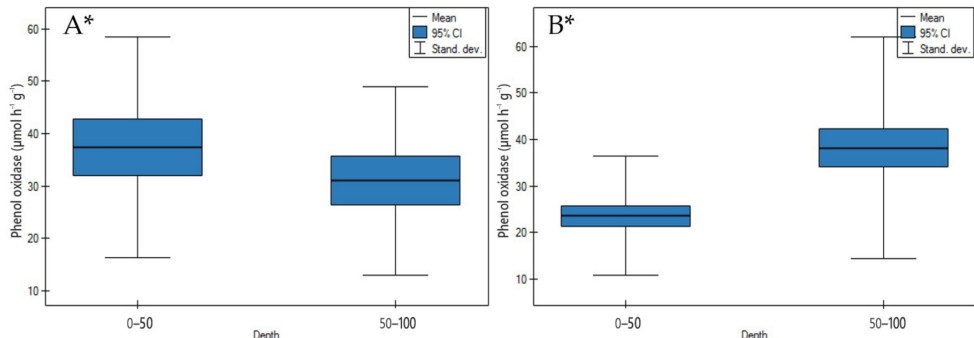

**Figure 14.** Phenol oxidase activity on: (**A**) undrained, (**B**) drained peatlands in 0–50 cm and 50–100 cm layers. * indicates statistically significant differences ($p < 0.05$) as revealed by ANOVA. (For all cases $n = 60$).

The highest decrease of enzyme activity in depth was measured for the Tagan Mire 1 (grasses) and the lowest for the Mukhrino fen peat (sedge woody, woody-cotton grass, sedge-*Sphagnum*, herbaceous-*Equisetum*) (Table S3).

In contrast to undrained peatlands, the drainage of the peatlands impacted on the significant increase of phenol oxidase activity in depth (Figure 14B, $p < 0.05$).

The highest rise in depth of phenol oxidase activity was found for the Zieleniec Mire (*Eriophoro-Sphagneti*) and the lowest in site 2 of the Września River valley (*Alneti*) (Table S4).

### 3.2.4. Peroxidase Activity

According to Jassey et al. [36], this enzyme catalyzes as an electron acceptor in the oxidation of polyphenols and aromatic amines in the presence of hydrogen peroxide and depolymerization of organic matter.

On undrained peatlands a significant decrease of peroxidase activity was shown with an increase in depth (Figure 15A, $p < 0.05$).

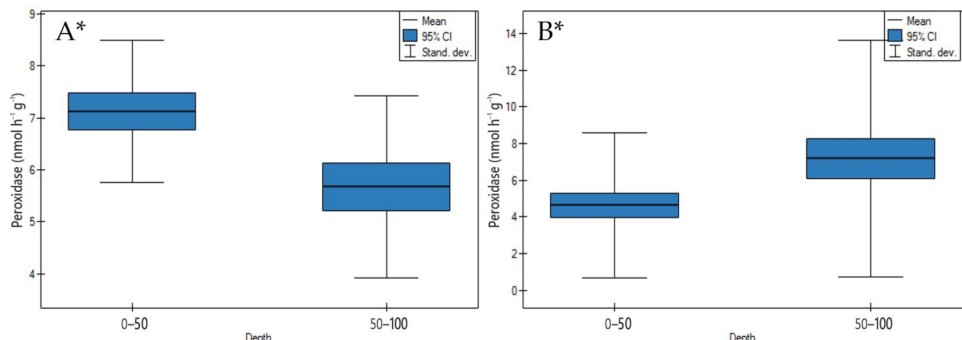

**Figure 15.** Peroxidase activity on: (**A**) undrained, (**B**) drained peatlands in 0–50 cm and 50–100 cm layers. * indicates statistically significant differences ($p < 0.05$) as revealed by ANOVA. (For all cases $n = 60$).

The highest decrease in depth of peroxidase activity was demonstrated for the Stążka Mire of Baltic-type (sedge-*Hypnum*, sedge, fragments of wood), while the lowest for the Mukhrino bog peat (*Sphagnum*) (Table S3).

In contrast to undrained ones, the drainage of the peatlands influenced the significant increase of peroxidase activity in depth (Figure 15B, $p < 0.05$).

The highest increase in depth of enzyme activity was measured in site 1 of the Września River valley (*Alneti*), and the lowest in site 1 of the General Dezydery Chłapowski Landscape Park (moorsh soil, alder swamp, wooden sedge, sedge-reed) (Table S4).

## 4. Discussion

### 4.1. Soil Physicochemical Properties

Undrained peatlands were characterized by a significant correlation coefficient ($r = 0.924$) between ash and bulk density values (Table 4). This indicates the importance of the mineral fraction in bulk density. This was related to a significant increase in these values in depth as compared to drained peatlands (Figures 3A and 4A). The drainage was the key reason for significant changes in ash and bulk density values in peatlands, which was shown by a significant correlation ($r = 0.990$) which was related to a significant increase in porosity in depth (Table 5, Figure 5A).

**Table 4.** Correlation coefficients between physicochemical and biochemical properties of undrained peatlands in 0–50 cm and 50–100 cm layers.

| Parameters | Moisture | Ash | Bulk Density | Porosity | $C_{HWE}$ | TOC | TN | $N\text{-}NH_4^+$ | $N\text{-}NO_3^-$ | C/N | Xanthine Oxidase | Urate Oxidase | Phenol Oxidase | Peroxidase |
|---|---|---|---|---|---|---|---|---|---|---|---|---|---|---|
| Moisture | - | | | | | | | | | | | | | |
| Ash | −0.174 | - | | | | | | | | | | | | |
| Bulk density | −0.248 | 0.924 | - | | | | | | | | | | | |
| Porosity | 0.049 | −0.238 | −0.217 | - | | | | | | | | | | |
| $C_{HWE}$ | −0.236 | −0.239 | −0.269 | 0.041 | - | | | | | | | | | |
| TOC | 0.266 | −0.323 | −0.161 | 0.050 | −0.588 | - | | | | | | | | |
| TN | −0.012 | −0.071 | −0.097 | 0.089 | −0.580 | 0.092 | - | | | | | | | |
| $N\text{-}NH_4^+$ | −0.269 | 0.250 | 0.138 | 0.001 | −0.023 | −0.603 | 0.657 | - | | | | | | |
| $N\text{-}NO_3^-$ | −0.261 | 0.316 | 0.139 | 0.011 | 0.210 | −0.749 | 0.252 | 0.809 | - | | | | | |
| C/N | 0.055 | 0.001 | 0.055 | −0.075 | 0.518 | 0.040 | −0.960 | −0.744 | −0.420 | - | | | | |
| Xanthine oxidase | 0.174 | −0.272 | −0.349 | 0.200 | −0.259 | −0.090 | 0.828 | 0.600 | 0.190 | −0.788 | - | | | |
| Urate oxidase | −0.330 | −0.966 | −0.873 | 0.204 | 0.964 | −0.364 | 0.789 | 0.114 | −0.887 | −0.814 | 0.954 | - | | |
| Phenol oxidase | 0.147 | −0.750 | −0.338 | 0.007 | −0.127 | 0.791 | −0.249 | −0.754 | −0.772 | 0.345 | −0.344 | 0.932 | - | |
| Peroxidase | −0.482 | 0.118 | 0.168 | −0.092 | 0.754 | −0.561 | −0.456 | 0.175 | 0.347 | 0.388 | −0.447 | 0.963 | −0.120 | - |

▧ significant correlation coefficient, $\alpha = 0.05$.

**Table 5.** Correlation coefficients between physicochemical and biochemical properties of drained peatlands in 0–50 cm and 50–100 cm layers.

| Parameters | Moisture | Ash | Bulk Density | Porosity | $C_{HWE}$ | TOC | TN | $N\text{-}NH_4^+$ | $N\text{-}NO_3^-$ | C/N | Xanthine Oxidase | Urate Oxidase | Phenol Oxidase | Peroxidase |
|---|---|---|---|---|---|---|---|---|---|---|---|---|---|---|
| Moisture | - | | | | | | | | | | | | | |
| Ash | −0.878 | - | | | | | | | | | | | | |
| Bulk density | −0.868 | 0.990 | - | | | | | | | | | | | |
| Porosity | 0.534 | −0.663 | −0.667 | - | | | | | | | | | | |
| $C_{HWE}$ | 0.428 | −0.500 | −0.499 | 0.285 | - | | | | | | | | | |
| TOC | 0.850 | −0.831 | −0.817 | 0.446 | 0.503 | - | | | | | | | | |
| TN | 0.091 | 0.076 | 0.100 | −0.179 | −0.378 | 0.181 | - | | | | | | | |
| $N\text{-}NH_4^+$ | 0.700 | −0.705 | −0.696 | 0.453 | 0.331 | 0.508 | 0.039 | - | | | | | | |
| $N\text{-}NO_3^-$ | 0.652 | −0.671 | −0.668 | 0.428 | 0.654 | 0.618 | −0.427 | 0.688 | - | | | | | |
| C/N | 0.587 | −0.689 | −0.699 | 0.506 | 0.690 | 0.539 | −0.681 | 0.462 | 0.860 | - | | | | |
| Xanthine oxidase | 0.531 | −0.554 | −0.553 | 0.423 | −0.016 | 0.208 | 0.098 | 0.833 | 0.309 | 0.210 | - | | | |
| Urate oxidase | −0.006 | −0.066 | −0.068 | 0.015 | −0.105 | 0.157 | 0.459 | −0.432 | −0.473 | −0.323 | −0.374 | - | | |
| Phenol oxidase | 0.587 | −0.694 | −0.702 | 0.710 | 0.559 | 0.742 | −0.622 | 0.450 | 0.839 | 0.949 | 0.222 | −0.314 | - | |
| Peroxidase | 0.656 | −0.755 | −0.766 | 0.739 | −0.725 | 0.580 | −0.486 | 0.589 | 0.787 | 0.912 | 0.389 | −0.182 | 0.874 | - |

■ significant correlation coefficient, α = 0.05.

Our results were in line with the study of Brandyk et al. [37], who showed that lowering of the water table starts the shrinkage of the upper layer, organic matter transformation, increasing of bulk density, and decreasing of total porosity values.

Moreover, Perdana et al. [38] pointed out that the moorsh-forming process caused an increase in the hydrophobic properties of peat. The volume shrinkage and the mass loss of peat contribute to it becoming more packed, resulting in increased bulk density values. This process may limit microbial activities and biochemical processes, which lead to the reduction of nutrient availability through the inhibition of organic matter mineralization [39].

The increase of moisture corresponded to higher ash and bulk density values in depth on drained peatlands (negative correlation r = −0.878 and r = −0.868, respectively) (Table 5).

On both undrained and drained peatlands, $C_{HWE}$ contents significantly decreased with an increase in depth (Figure 6A,B). This was due to the fact that $C_{HWE}$ is a mixture of low and high molecular weight compounds from plant root exudates, vegetation, and microbial enzymes, which determined their contents [40]. Our studies were in line with Jaszczyński et al. [41], who pointed out as a result of intense hydration of peat soils, oxygen access decreases, which is accompanied by a high water table and results in less leaching of $C_{HWE}$ to deeper layers of the profile. According to Szajdak and Szatyłowicz [42] and Woche et al. [43], the mechanism of the $C_{HWE}$ degradation depends on the aromaticity and complexity of dissolved organic matter molecules, whereas carbohydrates and amino acids increase the rate of this process.

A significant decrease of $C_{HWE}$ contents was accompanied by significant increase in TOC content on undrained and drained peatlands in increasing depth (Figure 7A,B). An increase in the depth of TOC content under saturated conditions may be related to a lower decomposition rate of organic matter. According to Klingenfuß et al. [44] on undrained peatlands, this was associated with the accumulation of humic compounds during the peat formation, where there occurs partial degradation of hemicelluloses, celluloses, and pectins.

The significant positive correlation (r = 0.850) indicated that higher TOC contents in the bottom layer on drained peatlands may be attributable to moisture values (Table 5). Our research did not confirm the effect of ash and bulk density values on TOC contents (negative correlation coefficient r = −0.831 and r = −0.817, respectively).

Based on our data, significantly the dynamics of changes in the relative of TN concentrations varied depending on the degree of decomposition on undrained peatlands (Table 1, Figure 8A). The reported nitrogen contents on drained peatlands were typical of drained organic soil where anaerobic conditions in the upper layers contribute to mineralization of the organic matter and a release of carbon into the atmosphere. Therefore nitrogen contents

oscillate before the protective effect of water and it may be secondarily accumulated in the peat-moorsh layers [45].

The significant rise of $N-NH_4^+$ concentrations in 0–50 cm as compared to that of the corresponding in the 50–100 cm layer on undrained peatlands, could be associated with an increase of nitrogen compounds ammonification and a reduction of the nitrification process in the upper layer (Figure 9A). This confirms the suggestion that ammonium ions adsorbed by soil colloids do not migrate deeper into the profile. Pawluczuk [46], Tripathi, and Sighn [47] showed that the intensity of the ammonification and nitrification processes are connected mainly with soil moisture, organic matter, degree of silting, land uses, and seasonal dynamics of released mineral forms of nitrogen.

Our study confirmed that lower ash values were related to negative correlation with $N-NH_4^+$ concentrations (r = −0.705) on drained peatlands (Table 5). Additionally, the significant positive correlation confirmed that moisture (r = 0.700) had an effect on $N-NH_4^+$ concentrations and it was associated with an increase of nitrogen compounds ammonification in the bottom layer (Table 5). On the other hand, Laine et al. [48] suggested that the lowering of the water table and changes in redox conditions stimulate nitrogen cycling and lead to an increase of ammonium formation as the product of nitrate reduction. It is difficult therefore to predict the transformation of nitrogen compounds concentrations with the functioning of different processes of the nitrogen cycle at the same time.

The drop of oxidation conditions in depth has led to a significant decrease of $N-NO_3^-$ concentrations on undrained peatlands suggesting a reduction of denitrification and nitrification processes (significant positive correlations with $N-NH_4^+$ concentrations, r = 0.809) in the bottom layers (Table 4, Figure 10A).

Significant differences of C/N ratios, which are indicators of organic matter mineralization, in layers between 0–50 and 50–100 cm were observed on undrained and drained peatlands (Figure 11A,B). The narrow C/N ratio (below 20) demonstrates a periodic drying out of soils and intensive mineralization. Our results were in line with the study of Krueger et al. [49], who pointed out that peat becomes exposed to aerobic conditions and stimulates organic matter mineralization. It led to the enrichment in nitrogen and explained a narrow C/N ratio in the upper as compared to the bottom layer on undrained peatlands.

Moreover, Szajdak et al. [26,50] suggested that the low C/N ratio in the upper layer leads to the emission of gaseous substances such as $CO_2$, $CH_4$, $N_2O$, and $N_2$ into the atmosphere. This is in accordance with the lower TOC contents and higher TN concentrations in the upper layer on undrained peatlands in our studies (Figures 7A and 8A).

However, the high C/N ratio in the deeper layer of the peat profile indicates more intensive accumulation of organic matter along with the rise of the degree of aromatic condensation and polycojugation in the molecules of HAs as compared to that in the upper layer [51].

### 4.2. Soil Biochemical Properties

#### 4.2.1. Xanthine Oxidase Activity

On undrained peatlands, a significant decrease (Figure 12A) and on drained peatlands, a significant increase (Figure 12B) of xanthine oxidase activity in depth were observed.

In this context, it was found that the significant rise of TN values (positive correlation r = 0.828) may stimulate higher xanthine oxidase activity in the upper, as compared to the bottom layer on undrained peatlands (Table 4).

This suggests that an increase of xanthine oxidase activity in the upper layer on undrained peatlands was the consequence of organic matter decomposition, labile compounds such as purine bases and peptides indicating the dominance of catabolic over anabolic processes. Peptides, purine bases, and aldehydes are fully degraded and more complex, perhaps partially oxidized by creating intermediate products. Formed compounds undergo polycondensation, creating macromolecules of humic substances [52].

The significant higher of xanthine oxidase activity in upper layers can be affected by root exudates of plants (Table 1). This may be explained by Gargallo-Garriga et al. [53] who

confirmed the pivotal role of root exudates in the rhizosphere and thus in plant-microbe-soil relationships. This is associated with a large variety of compounds released by plants into the rhizosphere, including low-molecular-weight primary metabolites (e.g., saccharides, amino acids, and organic acids), secondary metabolites (e.g., phenolics, flavonoids, and terpenoids), and inorganic molecules (e.g., carbon dioxide and water).

In contrast to undrained peatlands, on those drained significantly positive correlation (r = 0.833) between $N-NH_4^+$ and xanthine oxidase activity was important for the increase of this enzyme in the depth of the peat profile. In general, xanthine oxidase activity transformation is responsible for the changes in the structure of organic mass from hydrophilic to hydrophobic and influences the highest number of hydrocarbon chains and the lowest proportion of carboxyl groups [42].

The changes in chemical properties of transformed organic matter in peatlands created by a drop in the water table [54] had a significant influence on an increase of xanthine oxidase activity in depth as compared to undrained peatlands. Therefore, xanthine oxidase activity is considered as a good new moorsh-forming process indicator of oxidation and hydrological changes.

### 4.2.2. Urate Oxidase Activity

On undrained peatlands, a significant decrease (Figure 13A) and on drained peatlands, a significant increase (Figure 13B) of urate oxidase activity in depth were observed.

The significantly higher urate oxidase activity in the upper layer on undrained as compared to drained peatlands was accompanied by a significant higher $C_{HWE}$ and TN contents (positive correlation r = 0.964 and r = 0.789, respectively) (Table 4, Figure 13A). This can be connected with nitrogen heterocyclic compounds such as allantoin, which is a product of the catalytic decomposition of uric acid by urate oxidase activity [55,56]. This is reflected higher in the breakdown of purine derivatives and the formation of organic nitrogen compounds of low molecular weight, such as amino acids, amines, amides and amino sugars [56] on undrained than drained peatlands in the upper layer.

The significant negative correlation between urate oxidase activity and C/N (r = −0.814) may be attributable to more intensive mineralization in upper layers which influenced higher enzyme activity on undrained peatlands (Table 4). Our study has shown that the enzyme activity increased at lower ash and bulk density values (negative correlation coefficient r = −0.966, and r = −0.873, respectively) (Table 4). A significant positive correlation was found between urate, and xanthine oxidase activity (r = 0.954) which is also an essential enzyme in the ureide pathway and indicates the intensity of transformation of nitrogen compounds into peatlands (Table 4).

Additionally, Bobuľská et al. [57] and Li et al. [58] suggested that the high level of urate oxidase activity in the upper layer on undrained peatlands could have been caused by a change in the following factors, e.g., enzymes produced by microorganisms and roots exudates of higher-order plants.

On drained as opposite, to undrained peatlands, a significant increase of urate oxidase activity was considered as a consequence of a reduction in the water table and rise of the peat mineralization process (Figure 13B). Drainage of peat causes the denaturation of colloids and is therefore likely to be stimulated by a change in the properties of proteins from hydrophilic to hydrophobic [8], leading to a progressive increase of urate oxidase activity in deeper layers. This was confirmed by the study of Zhou et al. [59], who suggested that urate oxidase activities should be higher for a rise in oxygen concentration, which is a substrate of enzyme produced by bacteria in well-aerated conditions.

In this context, as reported by Urbanovà and Bàrta [60], long-term drainage has led to a strong decrease of decomposability and reduced microbial activity and it can be attributed to lower urate oxidase activity in the upper layers.

Hence, it may be stated that urate oxidase activity experiencing rapid changes with variations of environmental conditions can be used as a new moorsh-forming process indicator in association with the oscillation of the water table on drained peatlands.

### 4.2.3. Phenol Oxidase Activity

On undrained peatlands, a significant decrease (Figure 14A) and on drained peatlands, a significant increase (Figure 14B) of phenol oxidase activity in depth were observed.

In this context, significant higher phenol oxidase activity and negative correlation with ash values (r = −0.750) in the upper layers on undrained peatlands including free phenolics may be due to the humic substances with high molecular weight (including phenolic compounds) being particularly prone to precipitation and increased transformation through phenol oxidases activity under more aerobic conditions [61] (Table 4, Figure 14A).

The significant positive correlation between phenol and urate oxidase activity (r = 0.932) indicated that both these enzymes have the same potential in catalyzing oxidation and reduction reaction and transformation of organic matter in upper layers on undrained peatlands (Table 4).

On the other hand, during drainage conditions, a significant increase of porosity values (positive correlation r = 0.710) and TOC contents (positive correlation r = 0.742) in depth stimulated a higher phenol oxidase activity participating in the oxidation processes on drained as opposed to undrained peatlands (Table 5). This is associated with a significant negative correlation between phenol oxidase enzyme and bulk density values (r = −0.702) (Table 5). This contrast is responsible for the changes in the structure of organic mass constituting these soils, causing a modification of the properties of high molecular weight substances from hydrophilic to hydrophobic [62].

However, the high C/N ratio in the deeper layer of the peat profile was a significant positive correlation with phenol oxidase activity (r = 0.949) and indicated that more intensive accumulation of organic matter influenced the rise of enzyme activity (Table 5).

Our research results were in line with Brouns [63], who documented a positive effect with an increase in depth of phenol oxidase activity, implying that the enzyme is present even at great depths in anoxic peat conditions.

Studies confirmed that because of the fact that phenol oxidase activity is able to degrade lignin and soluble phenolic compounds, it can therefore be an important new moorsh-forming process indicator and could help monitor peat degradation during drainage.

### 4.2.4. Peroxidase Activity

On undrained peatlands, a significant decrease (Figure 15A) and on drained peatlands, a significant increase (Figure 15B) of peroxidase activity in depth were seen.

In this context, a rise of peroxidase activity in upper layers on undrained in contrast to drained peatlands may relate among others, to significantly higher $C_{HWE}$ contents (positive correlation r = 0.754) (Table 4, Figure 15A).

The significant positive correlation between peroxidase and urate oxidase activity (r = 0.963) indicated the same role of both enzymes in the overall process of organic matter humification and decomposition through catalyzing oxidation and reduction reaction (Table 4).

Jassey et al. [36] postulated that peroxidase activity, by causing the oxidation and transformation of total phenolic compounds, stimulates a partial or complete degradation and thereby influences carbon and nitrogen cycling in the peatland. Moreover, the same authors found that an increase of enzyme activity may be due to the rise of air temperatures above the upper layer. In contrast, Efremova and Ovchinnikova [64] suggested that a significant decrease of peroxidase depth activity is connected with natural factors such as high soil moisture, low temperature, and microbial activity.

In contrast to undrained peatlands, on those drained, besides the impacts of oxidation conditions, a significant rise in several factors, such as porosity values (positive correlation r = 0.739), availability of $N-NO_3^-$ concentrations (positive correlation r = 0.787), C/N (positive correlation r = 0.912), has led to a significant increase of peroxidase activity in depth of the peat profile (Table 5, Figure 15B).

Peroxidase activity has shown significant positive correlation with phenol oxidase (r = 0.874). As is well-known, the role of both enzymes in coupling reactions leading to polymerization is limited to the oxidation of the substrates (Table 5).

Our study results on drained peatlands were in line with the results of Tian et al. [65], who showed that this enzyme is closely related to soil organic matter transformations and suggested that it has too high a level of in situ oxidation conditions in the upper layer limited by peroxidase activity on drained peatlands. Moreover, Dec et al. [66] observed the influence of the long-term storage of carbon in soils and the biological availability of soil nitrogen on the accumulation of relatively stable peroxidases in the bottom layer. These variables suggested a higher peroxidase activity with the advancement of phenolic compounds transformation and the synthesis of macromolecules such as humic and fulvic acids in the bottom layer [67,68].

Peroxidase activity is the key index of the sustainability of soil organic matter and therefore can be potentially used as new moorsh-peat processes indicator.

## 5. Conclusions

On undrained peatlands, under saturation conditions, a significant decrease with an increase in depth of enzyme activities, xanthine oxidase, urate oxidase, phenol oxidase, and peroxidase, was observed. The significant correlation among urate and xanthine, phenol oxidase, peroxidase activity, having the same potential in catalyzing the oxidation and reduction reactions may confirm the increased effectiveness of their activity on undrained peatlands.

It was connected with significantly higher porosity values, $C_{HWE}$ and TN contents, $N\text{-}NH_4^+$ and $N\text{-}NO_3^-$ concentrations, and significantly lower ash and bulk density values in the upper layers.

In contrast to those undrained, on drained peatlands in oxidation conditions was demonstrated a significant increase in depth of enzyme activities: xanthine oxidase, urate oxidase, phenol oxidase, and peroxidase. The significant correlation between phenol oxidase and peroxidase activity involving the oxidation of high redox potential phenols confirmed the same mechanism of activity in catalyzing oxidation and reduction reactions.

On drained peatlands, in oxidation conditions significantly higher porosity values, TOC contents, TN concentrations, $N\text{-}NH_4^+$ concentrations, and C/N in the bottom layers were observed.

In this context, enzyme activities such as xanthine oxidase, urate oxidase, phenol oxidase, and peroxidase were found to be effective new indicators and tools for changes of the moorsh-forming process in relation to the oscillation of the water table caused by drainage of the peatlands.

**Supplementary Materials:** The following are available online at https://www.mdpi.com/2073-4395/11/1/113/s1. Table S1. Physicochemical properties of undrained peatlands: Mukhrino; Tagan Mire 1; Stążka Mire in 0–50 cm and 50–100 cm layers. Table S2. Physicochemical properties of drained peatlands: Wrześnica River valley; General Dezydery Chłapowski Landscape Park 1, 2, 3, 4; Tagan Mire; Great Batorowskie; Zieleniec Mire in 0–50 cm and 50–100 cm layers. Table S3. Oxidoreductive enzymes activity of undrained peatlands: Mukhrino; Tagan Mire 1; Stążka Mire in 0–50 cm and 50–100 cm layers. Table S4. Oxidoreductive enzymes activity of drained peatlands: Wrześnica River valley; General Dezydery Chłapowski Landscape Park 1, 2, 3, 4; Tagan Mire 2; Great Batorowskie; Zieleniec Mire in 0–50 cm and 50–100 cm layers.

**Author Contributions:** Conceptualization, L.W.S.; methodology, L.W.S.; writing—original draft preparation, L.W.S.; writing—review & editing, L.W.S.; supervision, L.W.S.; funding acquisition, L.W.S.; formal analysis, investigation, T.M.; formal analysis, M.S., T.M.; investigation, M.S., T.M.; statistical data calculations, M.S., T.M.; software, M.S. All authors have read and agreed to the published version of the manuscript.

**Funding:** This work was supported by the National Science Centre Poland [grants number 2013/09/B/NZ9/03169 and 2017/01/X/NZ9/00699]; the Russian Ministry of Education and Science [grant

number W 02.740.11.0325]; the RFFR, Peat—AcroCato [grant number W 09-05-00235], and the Interact, Transnational Access [grant number WP4-FP7].

**Conflicts of Interest:** The authors declare no conflict of interest.

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
