# Peer review of "Enzymatic Activity as New Moorsh-Forming Process Indicators of Peatlands"

_agronomy, doi:10.3390/agronomy11010113_

Round 1
Reviewer 1 Report
Authors have revised this manuscript extensively. The introduction has been reorganized and clearly indicated the knowledge gap as well as the objective of this paper.
There are few minor issues with the manuscript:
- It was mentioned that tested parameters among sites were analyzed by ANOVA. However, significant difference in tested parameters was represented due to depth within drained and undrained and not among sites.
- Table 1 and 2 need some formatting. Increase font size for axis titles. Indicate the significant difference due to depths using asterisk.
Reviewer 2 Report
Dear authors,
I studied the responses. I think the authors have done considerable work on revising their manuscript and gave satisfactory answers to the comments. However, there are some points that should be considered, in my opinion. I have attached a pdf file with all the commentaries.

Author Response
Please see the attachment

This manuscript is a resubmission of an earlier submission. The following is a list of the peer review reports and author responses from that submission.
Round 1
Reviewer 1 Report
The article titled as "New Moorsh-Forming Process Indicators" requires an extensive revision before publication. The study collected extensive data however the importance of that data is not clear in the introduction. The literature review needs to be reorganized to clearly show the knowledge gap. The objectives of the study were not clearly defined. The statistical analysis needs to be revised. Figures are not readable. Detailed discussion is required to highlight the significance of the study.
Reviewer 2 Report
Dear authors,
In this article, undrained and drained peatlands were studied. Soil samples were collected from two depths (0-50 cm and 50-100 cm). The authors analyzed soil physical and chemical parameters and enzyme activity and compared these values between depths by t-test. I think the authors have done considerable work in the literature review. However, the discussion seems to contains arbitrary observations that are not supported by the Results. In my opinion, the Discussion section needs to be revised and authors might consider the fact to add additional statistical analysis such as ANOVA or Discriminant Analysis.
Comments
L28-34. Please add references
L237. I think that the term “Soil physical, chemical and biochemical analyses” is more appropriate than “Analysis”
L257-272. Please add references for the experimental methods
L321. I suppose that a Student test has been done on each parameter to compare the two depths. Please, give more information such as why has been selected for this analysis, the null hypothesis, and the parameters that are used in this analysis. Also, did the authors applied the Levene test for equality of variance? Furthermore, the volume of data is quite large and the parameters are many, I would expect authors not to be restricted only to t-test analysis.
L325. I think that the use of “from … to…” is not appropriate as this term does not indicate the data range, (e.g. soil moisture was not 2% and went 6%, but it fluctuated between 2% and 6%). I believe that should be replaced with “between … and….” or “by….to….”. Please replace this phrase in all the Results section.
L338. Please replace “significant” with “significantly”
L483. In what frame is used the word “compared”? There is no statistical analysis that deals with it.
L513. How did the authors found that “the processes of TN mineralization were closely linked with…”?
L518. In the manuscript is written “significantly higher N-NH4 than N-NO3”, but there is no statistical analysis that supports this.
L542-543. Please add a reference.
L562-565, 604-607, 638-641, 664-667. Please revise the sentences, there is too much repetition.
L698. Why the increase of N-NO3 concentrations by depth increased significantly nitrate reductase activity, how the authors reached to this observation?
L715. In the manuscript is said “…the principal factor”, which are the rest factors? Are there?
Figure 2. What does Figure 2 serve? Does it give addition information? I think that Tables give all the necessary information.
Table 7. Why there are no data of Urate oxidase in Mukhrino and Tagan Mire 1 peatlands?